# Phylogenetic Clustering among Asylum Seekers with New HIV-1 Diagnoses in Montreal, QC, Canada

**DOI:** 10.3390/v13040601

**Published:** 2021-04-01

**Authors:** Hyejin Park, Bluma Brenner, Ruxandra-Ilinca Ibanescu, Joseph Cox, Karl Weiss, Marina B. Klein, Isabelle Hardy, Lavanya Narasiah, Michel Roger, Nadine Kronfli

**Affiliations:** 1Centre for Outcomes Research and Evaluation, Research Institute of the McGill University Health Centre, Montreal, QC H4A 3J1, Canada; hyejin.park@mail.mcgill.ca (H.P.); joseph.cox@mcgill.ca (J.C.); marina.klein@mcgill.ca (M.B.K.); 2McGill AIDS Centre, Lady Davis Institute, Jewish General Hospital, Montreal, QC H3T 1E2, Canada; bluma.brenner@mcgill.ca (B.B.); rIbanescu@jgh.mcgill.ca (R.-I.I.); 3Department of Medicine, Division of Infectious Diseases and Chronic Viral Illness Service, McGill University Health Centre, Montreal, QC H4A 3J1, Canada; 4Department of Epidemiology, Biostatistics and Occupational Health, McGill University, Montreal, QC H3A 0G4, Canada; 5Department of Medicine, Division of Infectious Diseases and Medical Microbiology, Jewish General Hospital, Montreal, QC H3T 1E2, Canada; karl.weiss@mcgill.ca; 6Department of Microbiology, Infectiology and Immunology, Université de Montréal, Montréal, QC H3T 1J4, Canada; isabelle.hardy.chum@ssss.gouv.qc.ca (I.H.); michel.roger.chum@ssss.gouv.qc.ca (M.R.); 7Centre de Recherche du Centre Hospitalier de l’Université de Montréal, Montréal, QC H2X 0A9, Canada; 8Direction Régionale de Santé Publique, CIUSSS Centre-Sud-de-l’Île-de-Montréal, Montréal, QC H2L 1M3, Canada; lavanya.narasiah.ccsmtl@ssss.gouv.qc.ca; 9Clinique des Réfugiés, CISSS Montérégie Centre, Brossard, QC J4Z 1A5, Canada

**Keywords:** HIV, phylogenetic analysis, phylogenetic clusters, transmission dynamics, migrants, asylum seekers

## Abstract

Migrants are at an increased risk of HIV acquisition. We aimed to use phylogenetics to characterize transmission clusters among newly-diagnosed asylum seekers and to understand the role of networks in local HIV transmission. Retrospective chart reviews of asylum seekers linked to HIV care between 1 June 2017 and 31 December 2018 at the McGill University Health Centre and the Jewish General Hospital in Montreal were performed. HIV-1 partial *pol* sequences were analyzed among study participants and individuals in the provincial genotyping database. Trees were reconstructed using MEGA10 neighbor-joining analysis. Clustering of linked viral sequences was based on a strong bootstrap support (>97%) and a short genetic distance (<0.01). Overall, 10,645 provincial sequences and 105 asylum seekers were included. A total of 13/105 participant sequences (12%; *n* = 7 males) formed part of eight clusters. Four clusters (two to three people) included only study participants (*n* = 9) and four clusters (two to three people) included four study participants clustered with six individuals from the provincial genotyping database. Six (75%) clusters were HIV subtype B. We identified the presence of HIV-1 phylogenetic clusters among asylum seekers and at a population-level. Our findings highlight the complementary role of cohort data and population-level genotypic surveillance to better characterize transmission clusters in Quebec.

## 1. Introduction

Many developed countries have witnessed recent shifts in their human immunodeficiency virus 1 (HIV-1) epidemics, with a disproportionate number of new HIV-1 diagnoses among migrant populations from countries of high HIV-1 endemicity [1,2,3]. Refugee claimants (i.e., asylum seekers) and sponsored refugees represent a growing proportion of people living with HIV-1 in Canada [4]. In 2018, migrants accounted for 40% of all new HIV-1 diagnoses in Canada [4] but only ~25% in 2015 [5]. A similar trend has been seen in Montreal, Quebec [6]. Certain migrants are at an increased risk of HIV acquisition both during and post-migration due primarily to the social disparities that accompany migration and changes in sexual behaviors that often occur post-migration [7].

Several studies, many using Bayesian methods, have shown significant post-migration HIV acquisition among certain migrant populations [2,8,9,10]. Timely HIV screening and linkage to care thus becomes particularly important for preventing both individual-level morbidity and mortality and onward transmission. However, delayed HIV screening and subsequent linkage to care have been reported among migrants with precarious status [1,7,11,12]; HIV-related stigma, limited knowledge of testing and treatment options, and fear of deportation are known to delay engagement along the HIV cascade of care among migrants [13,14].

Phylogenetic analysis, a form of molecular epidemiology, has been increasingly used to characterize HIV transmission and tailor potential prevention responses to better inform public health [15,16]. Recent studies using phylogenetic analysis found that small heterosexual clusters among migrants were commonly composed of people living with HIV from a single birth country [10,17]. Given the evolving epidemiology of HIV-1 in Canada, we aimed to use phylogenetics to characterize HIV-1 transmission clusters among newly-diagnosed asylum seekers and to better understand the role of networks in local HIV transmission. Here, we demonstrate phylogenetic clustering among asylum seekers and at a population-level. Our presented results highlight the potential complementary role of cohort data and population-level genotypic surveillance to identify transmission clusters among specific populations, while simultaneously upholding and promoting ethical conduct in HIV phylogenetic research.

## 2. Materials and Methods

### 2.1. Study Design, Setting, and Participants

The Chronic Viral Illness Service (CVIS) of the McGill University Health Centre (MUHC) and the Infectious Diseases Clinic (IDC) at the Jewish General Hospital (JGH) in Montreal, Quebec, Canada are multidisciplinary clinics focused on the care of individuals with HIV. In 2018, there were 1849 and 1135 patients who were in active care at the CVIS and the IDC, respectively, for either HIV infection alone or concomitant with another viral infection such as hepatitis B or C. The MUHC and JGH serve as the primary HIV referral centres for >75% of asylum seekers in Montreal.

All adult (18 years of age or older) patients living with HIV (with or without another viral co-infection) who were linked to HIV care at the CVIS or the IDC between 1 June 2017 and 31 December 2018 following a new diagnosis of HIV-1 and whose care was covered by the Interim Federal Health Program (i.e., asylum seekers) were included in this retrospective cohort study. A detailed description of the CVIS cohort has been described elsewhere [7]. Patients with HIV viral loads (VL) below 50 copies/mL (*n* = 12) did not have baseline HIV-1 genotypes and were thus excluded from analysis. The first genotype was analyzed among patients with more than one genotype (*n* = 17; 16%).

This study was approved by the Research Institute of the MUHC Research Ethics Board (MUHC 2020-5988). Ethics approval for encrypted and non-nominative anonymized phylogenetic surveillance of the HIV epidemic was obtained from the Ministre de la Santé and l’Institut national de santé publique du Québec with annual ethics approvals (CODIM-MBM-15-185) from the Integrated Health and Social Services University Network for West-Central Montreal.

### 2.2. Data Collection and Variables of Interest

The CVIS and IDC maintain clinical databases and electronic medical records, both of which were the sources of information for all variables of interest. Retrospective chart reviews using participants’ electronic medical records were used to collect baseline sociodemographic (age, sex, sexual orientation) and clinical (CD4 count and VL) characteristics, as well as HIV-1 genotypes. Authorization to access patient charts was obtained from the Director of Professional Services of the MUHC—Adult Sector.

### 2.3. Phylogenetic Analysis

HIV-1 genotyping was carried out as previously described to generate partial *pol* sequences, spanning the viral protease and the reverse transcriptase, HXB2 nucleotide positions 2253→2549 and 2550→3870, respectively [15,16]. All available HIV-1 *pol* sequences from the provincial genotyping database were aligned to consensus HXB2 sequences, removing gaps and cutting to identical sequence lengths (921 base pairs) using BioEdit. The provincial genotyping database (2002–2019) provides routine genotyping following treatment failure, as well as baseline genotyping of all HIV infected persons who live in Quebec, including students and recent migrants. The provincial genotyping database provides information regarding the date of sampling, sex, and the VL; in addition to this information, we obtained birth countries, dates of arrival in Canada, and CD4 counts at diagnosis for study participants.

Phylogenetic trees were constructed using neighbor-joining methods in MEGA10 software. Transmission clustering of linked viral sequences was inferred based on a strong bootstrap support (>97%, 1000 replicates) and a short genetic distance (<0.01). Genetic distance was computed using Kimura-2-parameter methods and in units of number of base units per site. The rate variation among sites was modeled with a gamma distribution (shape parameter = 1) [16]. Subtype B and non-B subtype trees were rooted against subtype K or N consensus sequences (accession numbers AJ24935 and AJ006022, respectively) to visually depict patterns of spread. A birthdate identifier was used to exclude replicate patient sampling within clusters. As such, clustering represents two or more members per cluster.

The HIV subtype was designated based on Stanford algorithms and was confirmed by REGA phylogenetic-based bootscan analysis. Natural amino acid polymorphisms and drug resistance mutations were assessed using the geno-2-pheno platform sequence and interpreted using the Stanford University HIV Drug Resistance Database [18].

Study cohort sequences (*n* = 105) were combined with de-identified first sequences of subtype B (*n* = 8947) or non-B subtype infections (*n*= 1746) from the provincial genotyping database to create population-level phylogenetic trees and to determine the presence of clusters at the provincial level [15]. To visualize clusters identified in the provincial subtype tree, a subtree combined subtype B clusters including Haitian asylum seekers with a subset of de-identified Haitian persons in the province (*n* = 132). The sequences were assigned a unique identifier, based on HIV-1 subtype, putative cluster group association, sex, and disease stage at first presentation. Clusters were characterized by sex, birth country, HIV risk population, and HIV subtype. GenBank accession numbers (MW788541 to MW788559) were obtained for asylum seekers belonging to the clusters.

### 2.4. Statistical Analyses

Summary statistics, medians, and interquartile ranges (IQR) or counts, and proportions were calculated to describe the sample. These analyses were performed in R-4.0.0.

## 3. Results

### 3.1. Patient Characteristics

A total of 105 asylum seekers were included, 83 from the MUHC and 22 from the JGH. Participant sociodemographic and clinical characteristics are presented in Table 1. The median age was 41 years, 51% were male, and 87% self-identified as heterosexual. Almost half (46%) of patients were of Haitian ethnicity, and approximately one-fifth (17%) were Nigerian. Half (50%) of the HIV viral subtypes were B.

The median CD4 cell count was 303 cells/μL with a viral load of 32,434 copies/mL at the time of diagnosis. A significant proportion was diagnosed late; 62% were late presenters (CD4 < 350 cells/μL) and 30% had advanced HIV infection (CD4 < 200 cells/μL). One-fifth (20%) presented with high-level viremia (VL > 100,000 copies/mL) and five (5%) patients had VLs > 500,000 copies/mL. Opportunistic infections were rare; one newly-diagnosed asylum seeker at the MUHC was diagnosed with cerebral toxoplasmosis.

### 3.2. Transmission Clusters

Overall, 105 sequences from study participants were combined with 8947 subtype B infections and 1746 non-B subtype infections genotyped in the province (2002–2019). Of these, a total of 13/105 participant sequences (12%; *n* = 7 males (54%)) formed part of eight phylogenetic clusters (Table 2). Among these, four clusters (cluster sizes = 2–3 people) included study participants alone (*n* = 9), whereas another four clusters (cluster sizes = 2–3 people) included four study participants clustered with six individuals from the provincial genotyping database. Among the former four clusters, three cluster pairs (CT270, CAG41, G5) were among known Haitian or Nigerian heterosexual couples, while cluster C636 represented a Haitian subtype-B cluster with two males and one female (none of whom were known partners). Among the four participant–provincial genotyping database clusters, these included one heterosexual Haitian female–provincial genotyping database male pair (CT268) and a female from Saint Vincent clustered with one provincial genotyping database male and one provincial genotyping database female (C569). In addition, there was two male–male clusters, one with a Haitian male clustered with two males from the provincial genotyping database (C638) and one pair consisting of an Algerian male clustered with a male from the provincial genotyping database (C645). Of the eight clusters, five clusters consisted of pairs (cluster size = 2). Among the eight clusters, resistance mutations were similar among cluster participants except for cluster G5.

There were seven individuals from Haiti in four subtype-B clusters (Figure 1). Given the large number of subtype B infections in the province (*n* = 8947), we combined the 48 subtype B sequences from Haitian asylum seekers with 132 de-identified Haitian patients to visually depict patterns of clustering. Overall, most new diagnoses in Haitian groups represent singleton transmissions or small cluster networks.

## 4. Discussion

Our study retrospectively identified the presence of HIV-1 phylogenetic clusters, among both asylum seekers—most of whom were known heterosexual couples—and asylum seekers and HIV-infected individuals from the provincial genotyping database. Almost all of our clustered study participants presented with late HIV-1 infection (CD4 count < 350 cells/μL), suggesting that HIV acquisition likely occurred prior to arrival in Canada. In fact, all genotypes were performed within six months of participants’ arrival, further underscoring the likelihood of HIV acquisition prior to arrival in Canada. A post-hoc review of participants’ travel trajectories failed to reveal extended lengths of stay in other countries following departure, suggesting that HIV acquisition not only occurred prior to arrival in Canada but likely in participants’ birth countries. If this is the case, our study highlights that routine HIV testing may be sub-optimal in developing countries [19,20] despite global efforts to diagnose 90% of those infected with HIV [21]. Furthermore, while our results suggest remote transmission, domestic HIV transmission cannot be excluded. Although imperfect, molecular surveillance is increasingly being used to identify and respond to HIV transmission clusters to end HIV epidemics [22]. Our findings thus demonstrate the potential complementary role of cohort data and population-level genotypic surveillance to identify transmission clusters among specific populations.

Our understanding of the role of networks in local HIV transmission is limited by data collected in our provincial genotyping database. While provincial data excluded key demographic and clinical information that could have substantiated some of our hypotheses, their deliberate exclusion mitigates some serious ethical concerns to individuals and communities that may arise from phylogenetic research [23]. Identification of transmission clusters may further stigmatize and marginalize groups vulnerable to HIV such as certain migrant populations. Further, phylogenetic analysis could contribute evidence to inferences about specific transmission events between individuals—data that could potentially support HIV criminalization prosecutions in countries where this occurs. While British Columbia has opted to use an automated, near real-time system to identify HIV transmission hotspots [23,24], there remains a critical gap between this approach and its overall value to public health [25].

Our study has several policy and care implications. All asylum seekers entering Canada must undergo an immigration medical examination (performed within 30 days from arrival), which includes mandatory HIV screening. To improve public health surveillance, genotypes could be performed automatically (i.e., reflexed) during screening, and provincial transmission dynamics could be subsequentially analyzed to ensure timely prevention, linkage to care, and treatment initiation in any identified subpopulation. Those who test negative should be counselled on risk for HIV acquisition and offered pre-exposure prophylaxis if appropriate. Furthermore, we witnessed clustering among individuals from similar birth countries (i.e., ethnocultural clustering), supporting the importance of culturally-appropriate HIV prevention and treatment services. As we were unable to exclude the possibility of domestic HIV transmission, point-of-care HIV testing at the time of entry or as part of the immigration medical exam could be considered. Such test-and-treat models of care would mitigate individual-level morbidity and mortality and prevent onward transmission. In addition, as at-risk sexual behaviors often change following an HIV diagnosis [26], ensuring both timely linkage to care and monitoring of transmission clusters becomes even more relevant from a public health standpoint. Future studies should seek community engagement in the research process given the sensitivity of phylogenetic data [23], with the goal of contributing to public health surveillance, informing public health policy, but simultaneously promoting ethical conduct of HIV phylogenetic research.

Our study has several limitations. Phylogenetic clustering implies closely related infection; however, it does not imply direct transmission. It is possible that there were other infections in the transmission networks that have yet to be diagnosed, were diagnosed elsewhere, or did not have available *pol* sequences. Furthermore, it was not possible to describe the possible transmission networks of those who were not in phylogenetic clusters. Third, while CD4 count and VL data were used to approximate duration of HIV infection, determining the recency of infection using validated methods (e.g., genetic sequence diversity) [27,28,29] may have provided complimentary information to estimate the timing of HIV acquisition. Finally, social desirability bias may have impacted the reporting of sexual orientation, particularly among MSM [10].

Our study has highlighted the potential complementary role of cohort data and population-level genotypic surveillance to better characterize HIV transmission clusters in Quebec, while simultaneously upholding and promoting ethical conduct in HIV phylogenetic research.

## Figures and Tables

**Figure 1 viruses-13-00601-f001:**
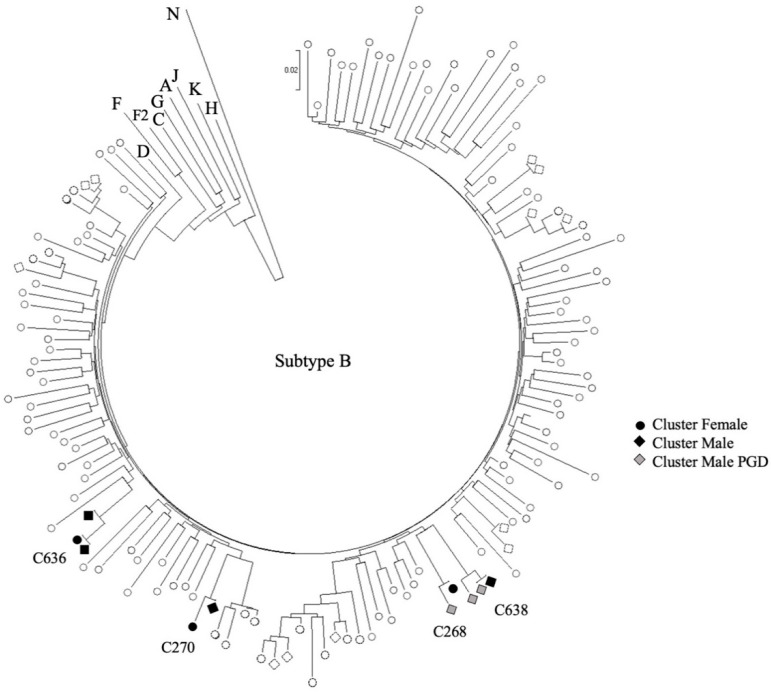
Phylogenetic tree of HIV-1 subtype B using a subsample of *pol* sequences from de-identified individuals (infected Haitians (*n* = 132) in the province) from Table 1. Clustering was inferred based on bootstrap support (>97%) and short genetic distance (<0.01). PGD: Provincial genotyping database.

**Table 1 viruses-13-00601-t001:** Baseline characteristics of the study sample.

	Overall (*n* = 105)	MUHC (*n* = 83)	JGH (*n* = 22)
Age (median (IQR))	41 [35;47]	40 [35;46]	44 [39;51]
Sex			
Female	51 (49%)	46 (55%)	5 (23%)
Sexual orientation			
Heterosexual	91 (87%)	70 (84%)	21 (95%)
LGBTQ	14 (13%)	13 (16%)	1 (5%)
Birth country			
Africa			
Burundi	4 (4%)	2 (2%)	2 (9%)
DRC	6 (6%)	5 (6%)	1 (4%)
Nigeria	18 (17%)	18 (22%)	0 (0%)
Other	23 (22%)	20 (24%)	3 (14%)
Americas			
Haiti	48 (45%)	36 (44%)	12 (55%)
Other	6 (6%)	2 (2%)	4 (18%)
HIV subtype			
AG	19 (18%)	17 (21%)	2 (9%)
B	52 (50%)	37 (45%)	15 (68%)
C	14 (13%)	12 (14%)	2 (9%)
G	6 (6%)	6 (7%)	0 (0%)
Other	14 (13%)	11 (13%)	3 (14%)
CD4 at diagnosis, cells/μL (median; range; (IQR))	303; 7–1567; [181;416]	308; 7–811; [194;392]	221; 29–1567; [159;474]
CD4 < 200	31 (30%)	21 (25%)	10 (45%)
CD4 200–350	34 (32%)	29 (35%)	5 (23%)
CD4 > 350	40 (38%)	33 (40%)	7 (32%)
VL at diagnosis, copies/mL (median; range; (IQR))	32,434; <20–1,348,292; [7429;80,017]	32,434; <20–1,348,292; [6962;95,958]	33,913; 89–347,068; [11,244;73,649]
VL 0–99,999	84 (80%)	65 (78%)	19 (86%)
VL 100,000–499,999	16 (15%)	13 (16%)	3 (14%)
VL ≥ 500,000	5 (5%)	5 (6%)	0 (0%)

DRC: Democratic Republic of the Congo; IQR: interquartile range; JGH: Jewish General Hospital, LGBTQ: lesbian, gay, bisexual, transgender, and/or queer; MUHC: McGill University Health Centre; VL: viral load.

**Table 2 viruses-13-00601-t002:** Individual cluster characteristics among the study sample.

Cluster Identification	HIV Subtype	Birth Country	Sex	Self Reported Sexual Orientation	Date of Arrival in Canada	Date of Sampling	CD4 at Genotype (cells/μL)	VL at Genotype (copies/mL)	Resistance Mutations
Clusters with Study Participants Only									
C636.01	B	Haiti	M	HET	08/2017	10/2017	167	32,908	K103N
C636.02		Haiti	M	HET	10/2017	11/2017	347	99,274	K103N
C636.03		Haiti	F	HET	09/2017	12/2017	687	19,896	K103N
CT270.01	B	Haiti	M	HET	09/2017	11/2017	209	2549	WT
CT270.02		Haiti	F	HET	09/2017	11/2017	191	68,743	WT
CAG41.01	AG	Nigeria	M	HET	12/2017	04/2018	333	181,853	WT
CAG41.02		Nigeria	F	HET	12/2017	04/2018	316	11,524	WT
G5.01	G	Nigeria	M	HET	12/2017	02/2018	11	112,947	M184V, T215F, G190A
G5.02		Nigeria	F	HET	12/2017	02/2018	531	40,988	WT
**Clusters with Study Participants and Individuals from the Provincial Genotyping Database**									
CT268.01	B	N/A	M	N/A	N/A	08/2017	N/A	1628	K103KN
CT268.02		Haiti	F	HET	04/2018	04/2018	382	5487	K103KN
C569.01	B	N/A	M	N/A	N/A	08/2009	N/A	12,158	WT
C569.02		N/A	F	N/A	N/A	03/2016	N/A	959	WT
C569.03		Saint Vincent	F	HET	06/2018	06/2018	122	15,752	WT
C638.01	B	Haiti	M	MSM	08/2017	11/2017	71	1,154,977	WT
C638.02		N/A	M	N/A	N/A	02/2018	N/A	225,274	WT
C638.03		N/A	M	N/A	N/A	02/2018	N/A	238,744	WT
C645.01	B	N/A	M	N/A	N/A	02/2011	N/A	385,982	WT
C645.02		Algeria	M	MSM	11/2017	06/2018	279	460,002	WT

F: female; HET: heterosexual; M: male; MSM: men who have sex with men; N/A: not available; VL: viral load; WT: wild type.

## Data Availability

The data presented in this study are available on request from the corresponding author. Sequence data of phylogenetic clusters can be found at GenBank using accession numbers MW788541 to MW788559.

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
