# Peer review of "Phylogenetic Clustering among Asylum Seekers with New HIV-1 Diagnoses in Montreal, QC, Canada"

_viruses, 2021, doi:10.3390/v13040601_

Round 1
Reviewer 1 Report
In this manuscript Hyejin Park et al. describe a relevant phylogeny-based study of HIV-1 infections in asylum seekers in Canada. The data could be further analysed by querying public databases and performing Bayesian evolutionary and phylogeographic analyses. These would likely allow complementing the results and provide a better understanding on the origin of the reported infections and possible ongoing transmission chains. Nonetheless, the work provides valuable information on the studied group and has relevant policy and health-care implications for the country. Thus, I recommend that the paper is accepted after the following revisions:
- The studied HIV-1 sequences and anonymized data on the time and place of sequencing should be deposited in public databases. The IDs from the deposited data should be included in the manuscript. This is central for future studies related to phylogenetic and phylogeographic evolutionary analysis of HIV-1 sequences.
- When referring specifically to HIV-1 the term HIV should be replaced by HIV-1. I also consider good practice that the first occurrence of the term in the introduction defines the abbreviation (Human immunodeficiency virus 1 (HIV-1)).
- (Page 3, line 123) Give more details should be given on the criteria considered for clustering. Regarding genetic distance was it considered the mean distance between all sequences or the maximum pairwise distance? it would be better to include the distance in substitutions per site in addition to % to facilitate comparison with the criteria used in other studies. Were pairs of sequences a cluster or only >2 sequences?
- (Page 3, line 121) Indicate what was the substitution model used in maximum likelihood phylogenetic analysis and how was it determined to be the best fitting substitution model for this analysis.
- (Page 3, line 124) “trees were rooted against subtype K or N consensus”. Indicate the accession numbers of the sequences used to root the trees and explain the rational for the use subtype K or N consensus.
- (Page 3, line 128) Clarify how were the cohort sequences combined with the "provincial genotyping database" sequences? How many sequences from the study cohort and how many sequences from the database were used? how was the database queried? Was BLAST or other similar search method used to identify the sequences from the database with the closest genetic proximity? If so, what were the search parameters and query used (the available partial pol sequence and/or partitions in protease and transcriptase coding sequences).
- (Page 3, line 132) What was the method used to define the HIV-1 subtype? There are different approaches to perform HIV-1 subtyping, including: similarity-based (e.g. Stanford) or phylogenetic-based (e.g. REGA, SCUEAL or SNAPPy). Phylogenetic-based tools are considered the most sensitive and specific. Thus, results obtained with similarity-based methods should be confirmed with phylogenetic methods.
- (Page 3, line 134) The methods describing the statistical analysis do not provide enough detail to reproduce the performed analysis.
- (Page 6, line 174) Give more details on figure 1 legend. Is it a representation of the Maximum likelihood tree? If 10,645 sequences were included in the phylogenetic analysis (as indicated in page 4, line 153) why only some sequences and clusters are shown in the figure? What were the criteria used for the selection of this subset of sequences?
- (Page 6, line 181) The fact that some sequences from studied participants cluster with individuals from the provincial genotyping database (table 2) could be considered evidence against the hypothesis that "HIV acquisition likely occurred prior to arrival in Canada" (page 6, line 181). Is it likely that these individuals from the provincial genotyping database (i.e. CT268.01, C569.01, C569.02...) were also migrants infected before arrival to Canada? This hypothesis could had been further studied by querying public databases to investigate if sequences clustering with the ones from the studied participants could be found and what were the countries where these were previously sampled.
Author Response
Please see the attachment. Thank you.
Response to Reviewer 1 Comments
Point 1: The studied HIV-1 sequences and anonymized data on the time and place of sequencing should be deposited in public databases. The IDs from the deposited data should be included in the manuscript. This is central for future studies related to phylogenetic and phylogeographic evolutionary analysis of HIV-1 sequences.

Response 1: The GenBank Accession numbers were obtained for sequences of asylum seekers belonging to clusters.
(Page 3, line 151-152) “GenBank accession numbers (MW788541 to MW788559) were obtained for asylum seekers belonging to clusters.”
Point 2: When referring specifically to HIV-1 the term HIV should be replaced by HIV-1. I also consider good practice that the first occurrence of the term in the introduction defines the abbreviation (Human immunodeficiency virus 1 (HIV-1)).
Response 2: The manuscript has been modified as suggested.
Point 3: (Page 3, line 123) Give more details should be given on the criteria considered for clustering. Regarding genetic distance was it considered the mean distance between all sequences or the maximum pairwise distance? it would be better to include the distance in substitutions per site in addition to % to facilitate comparison with the criteria used in other studies. Were pairs of sequences a cluster or only >2 sequences?
Response 3: This paragraph (page 3, lines 129-138) was modified to address the reviewer’s comment: “Phylogenetic trees were constructed using neighbor-joining methods in MEGA10 software. Transmission clustering of linked viral sequences was inferred based on strong bootstrap support (>97%, 1000 replicates) and short genetic distance (<0.01). Genetic distance was computed using Kimura-2-parameter methods and in units of number of base units per site. The rate variation among sites was modeled with a gamma distri-bution (shape parameter =1) [16]. Subtype B and non-B subtype trees were rooted against subtype K or N consensus sequences (Accession numbers AJ24935 and AJ006022, re-spectively) to visually depict patterns of spread. A birthdate identifier was used to exclude replicate patient sampling within clusters. As such, clustering represents two or more members per cluster.”
The abstract was also modified to replicate the changes made to the manuscript.
Point 4: (Page 3, line 121) Indicate what was the substitution model used in maximum likelihood phylogenetic analysis and how was it determined to be the best fitting substitution model for this analysis.
Response 4: This comment was addressed in Response 3.
Point 5: (Page 3, line 124) “trees were rooted against subtype K or N consensus”. Indicate the accession numbers of the sequences used to root the trees and explain the rational for the use subtype K or N consensus.
Response 5: Page 3, lines 134-135 was modified to address this comment: “Subtype B and non-B subtype trees were rooted against subtype K or N consensus sequences (Accession numbers AJ24935 and AJ006022, respectively).”
In general, a phylogenetic tree is rooted to the common ancestor from which species are generated to gives an indication of the directionality of evolutionary change. For HIV, there is no evident directionality in the tree. We added a subtype N or K sequences to our phylogenetic analysis so that we were able to root the tree to either sequence for the visual aspect of the tree. The methods were not expanded to include this.
Point 6: (Page 3, line 128) Clarify how were the cohort sequences combined with the "provincial genotyping database" sequences? How many sequences from the study cohort and how many sequences from the database were used? how was the database queried? Was BLAST or other similar search method used to identify the sequences from the database with the closest genetic proximity? If so, what were the search parameters and query used (the available partial pol sequence and/or partitions in protease and transcriptase coding sequences)
Response 6:
(Page 3, lines 143-148) now reads:
“Study cohort sequences (n=105) were combined with de-identified first genotypes of subtype B (n=8,947) or non-B subtype infections (n= 1,746) from the provincial genotyping database to create population-level phylogenetic trees and to determine the presence of clusters at the provincial level [15]. To visualize clusters identified in the provincial sub-type tree, a subtree combined subtype B clusters including Haitian asylum seekers with a subset of de-identified Haitian persons in the province (n=132).”
The provincial database was not queried or BLASTed. Samples were not queried using BLAST since there are too few samples from Haiti in the GenBank database. GenBank accession numbers were obtained for asylum seekers within clusters (MW788541 to MW788559). Dr. Bluma Brenner (co-investigator) started the database when the provincial genotyping program was initiated. No additional information was added to the manuscript as a result.
Point 7: (Page 3, line 132) What was the method used to define the HIV-1 subtype? There are different approaches to perform HIV-1 subtyping, including: similarity-based (e.g. Stanford) or phylogenetic-based (e.g. REGA, SCUEAL or SNAPPy). Phylogenetic-based tools are considered the most sensitive and specific. Thus, results obtained with similarity-based methods should be confirmed with phylogenetic methods.
Response 7: (Page 3, line 139-140) now reads: “HIV subtype was designated based on Stanford algorithms and confirmed by REGA phylogenetic-based bootscan analysis.”
Point 8: (Page 3, line 134) The methods describing the statistical analysis do not provide enough detail to reproduce the performed analysis.
Response 8: The descriptive analyses in Table 1 can be reproduced using the statistical analyses provided, therefore no changes were made to the manuscript.
Point 9: (Page 6, line 174) Give more details on figure 1 legend. Is it a representation of the Maximum likelihood tree? If 10,645 sequences were included in the phylogenetic analysis (as indicated in page 4, line 153) why only some sequences and clusters are shown in the figure? What were the criteria used for the selection of this subset of sequences?
Response 9: We have added additional details to Figure 1 in the manuscript:
- (Page 3, lines 146-148): “To visualize clusters identified in the provincial subtype tree, a subtree combined subtype B clusters including Haitian asylum seekers with a subset of de-identified Haitian persons in the province (n=132).” and
- (Page 5, lines 197-201): “Given the large number of subtype B infections in the province (n=8947), we combined the 48 subtype B sequences from Haitian asylum seekers with 132 de-identified Haitian patients, to visually depict patterns of clustering. Overall, most new diagnoses in Haitian groups represent singleton transmissions or small cluster networks.”
The legend (page 6, lines 209-213) has also been modified to state: “Figure 1. Phylogenetic tree of HIV-1 subtype B using a subsample of pol sequences from de-identified individuals from the provincial genotyping database and study participants born in Haiti. Four subtype B clusters of Haitian study participants are depicted with a subset of HIV-1 infected Haitians (n=132) in the province. Clustering was inferred based on bootstrap support (>97%) and short genetic distance (<0.01).”
Point 10: (Page 6, line 181) The fact that some sequences from studied participants cluster with individuals from the provincial genotyping database (table 2) could be considered evidence against the hypothesis that "HIV acquisition likely occurred prior to arrival in Canada" (page 6, line 181). Is it likely that these individuals from the provincial genotyping database (i.e. CT268.01, C569.01, C569.02...) were also migrants infected before arrival to Canada? This hypothesis could had been further studied by querying public databases to investigate if sequences clustering with the ones from the studied participants could be found and what were the countries where these were previously sampled.
Response 10: As information related to birth countries was not available for individuals in the provincial genotyping database, we cannot infer they are migrants nor comment on their likelihood to have acquired HIV-1 prior to their arrival in Canada. Querying public databases was outside the scope of our study.
Reviewer 2 Report
This is a very well written and executed study with an important public health message. I have very few comments and some suggestions.
Please describe how many clusters were found that did not include asylum seekers – are they more or less likely to be in a cluster?
Suggestions (please feel free to disregard):
1) As additional background data, maybe add publicly available sequences from home countries to your dataset to see if you’ll be able to identify more clusters.
2) If you could analyse all sequences from patients with multiple sequences (using one at a time in the analysis) – interesting to see if same clusters would still meet criteria for clustering (assuming any of the clustered individuals actually have more than 1 sequence).
Minor:
- Line 97 – Give the number (%) of such patients.
- Please describe what provincial genotyping database is – what geographic representation? Duplicates from the study cohort? Only sequences from Canadian residents?
- Please specify substitution model used to build trees
- Figure 1 – add colour to represent country
Author Response
Please see the attachment. Thank you.
Response to Reviewer 2 Comments
Point 1: Please describe how many clusters were found that did not include asylum seekers – are they more or less likely to be in a cluster?
Response 1: We only investigated phylogenetic clusters involving asylum seekers, therefore there were no clusters that did not include asylum seekers.
Point 2: Suggestions (please feel free to disregard):
1) As additional background data, maybe add publicly available sequences from home countries to your dataset to see if you’ll be able to identify more clusters.
2) If you could analyse all sequences from patients with multiple sequences (using one at a time in the analysis) – interesting to see if same clusters would still meet criteria for clustering (assuming any of the clustered individuals actually have more than 1 sequence).
Response 2:
- We did not have access to publicly available sequences from participants’ home countries.
- Page 3, Lines 134-138 now includes the following: “A birthdate identifier was used to exclude replicate patient sampling within clusters. As such, clustering represents two or more members per cluster.”
Point 3: Line 97 – Give the number (%) of such patients.
Response 3: The proportion (16%) has been included in line 102.
Point 4: Please describe what provincial genotyping database is – what geographic representation? Duplicates from the study cohort? Only sequences from Canadian residents?
Response 4: We have added the following to the first paragraph of section 2.3 (page 3, lines 122-125): “The provincial genotyping database provides routine genotyping following treatment failure, as well as baseline genotyping of all HIV infected persons who live in Quebec, including students and recent migrants.”
Point 5: Please specify substitution model used to build trees
Response 5: We have expanded our Methods (page 3, lines 129-134) to include: “Phylogenetic trees were constructed using neighbor-joining methods in MEGA10 software. Transmission clustering of linked viral sequences was inferred based on strong bootstrap support (>97%, 1000 replicates) and short genetic distance (<0.01). Genetic distance was computed using Kimura-2-parameter methods and in units of number of base units per site. The rate variation among sites was modeled with a gamma distribution (shape parameter =1).”
Point 6: Figure 1 – add colour to represent country.
Response 6: There is one unique country in Figure 1, Haiti, therefore no colour was added.
Reviewer 3 Report
The authors of this study have performed retrospective epidemiological and phylogenetic cluster analyses of a cohort of HIV-infected asylum seekers using databases from McGill University Health Centre and the Jewish General Hospital in Montreal between June 2017 and Dec 2018. The conclusions of this work are significant and suggest that among the genetic clusters that HIV subtype B was the predominant clade represented, and that immigrants from the same birth country typically shared the same phylogenetic cluster and therapy-resistance mutations which implies pre-immigration viral transmission, as opposed to post-migration HIV-1- acquisition. This is a well-controlled and clinically significant epidemiological study which describes the patterns and routes of HIV transmission among asylum seekers within a defined population over a specific time period.
Author Response
Thank you.